# Description and Complications of a New Modified Semi-Closed Castration Technique in Horses

**DOI:** 10.3390/vetsci12080720

**Published:** 2025-07-31

**Authors:** Marco Gandini, Cristina Bertone, Gessica Giusto

**Affiliations:** Department of Veterinary Sciences, University of Turin,10095 Grugliasco, Italy; cristina.bertone@unito.it (C.B.); gessica.giusto@unito.it (G.G.)

**Keywords:** semi-closed technique, castration, equine

## Abstract

This study presents a new surgical method for castrating male horses, ponies, and donkeys that aims to reduce the risk of complications after surgery. Eighty-five animals were treated using this technique in a hospital setting under general anesthesia. The new method involves making a small cut near the groin area, carefully inspecting the tissues, and securely closing the area after removing the testicles. This careful approach led to a very low complication rate and helped the animals recover quickly and return to normal activities. Compared to traditional techniques, this method appears safer, with fewer risks of bleeding, infections, or organs accidentally slipping out through the surgical site. The authors also suggest a simpler way to describe castration methods, based on whether the inner tissue layer is left open or closed after surgery. Overall, this new technique could lead to safer and faster recoveries for horses undergoing castration.

## 1. Introduction

Equine castration remains one of the most frequently surgical procedures performed in veterinary practice, widely undertaken for managing aggressive or unwanted stallion behaviors, preventing undesired breeding, and addressing certain medical conditions such as inguinal herniation or testicular neoplasia [1]. Although widely regarded as a routine procedure, castration carries the potential for a wide array of postoperative complications. These range from minor, self-limiting issues such as scrotal edema and localized infections to serious, potentially life-threatening events including hemorrhage, eventration, and peritonitis [1,2]. Several surgical approaches have been described and are typically classified as open, closed, and semi-closed techniques. The open technique, characterized by leaving the parietal vaginal tunic open, provides clear visualization and hemostasis of the spermatic cord but carries a higher risk of complications such as infection and evisceration [1]. Closed and semi-closed techniques, which involve partial or complete closure of the vaginal tunic, potentially reduce these risks by preventing direct communication between the abdominal cavity and external environment [2,3]. Specifically, semi-closed castration techniques have been shown to lower the risks of evisceration and infection compared to open approaches, largely due to the partial retention and closure of the parietal vaginal tunic [3,4]. Recent advancements and modifications in castration techniques aim to reduce the incidence of complications and enhance postoperative recovery. Techniques such as minimally invasive compartmentalized modified open castration, which integrates minimal scrotal dissection with primary closure, have reported low complication rates and rapid return to normal function [5]. Furthermore, closed inguinal approaches, performed under optimal surgical conditions, have demonstrated significantly lower postoperative inflammatory responses compared to traditional field castrations, as indicated by substantially reduced Serum Amyloid A levels [6]. Despite these advancements, variability in perioperative management, surgical settings, and surgeon expertise continues to influence complication rates significantly [7]. Studies underscore the importance of meticulous surgical technique, controlled aseptic conditions, and precise ligation methods to minimize postoperative morbidity [8,9,10]. This study introduces a novel semi-closed castration technique, aiming to further reduce postoperative complications, particularly hemorrhage and infection, and enhance overall recovery compared to established open and closed methods. By integrating the advantages of minimally invasive approaches with the safety of closed vaginal tunic management, this technique may offer a significant improvement in equine surgical castration outcomes.

## 2. Materials and Methods

### 2.1. Animals

Eighty-five sexually intact adult male equids—including horses, ponies, and donkeys—presented to the Veterinary Teaching Hospital for elective castration between 2015 and 2025 were included in this study. Informed consent was obtained from all owners prior to surgery. Institutional ethics committee approval was not sought, as the procedure represented a routine clinical intervention performed by the authors, and the novel technique represented a refinement of previously established surgical methods. Eligibility criteria required that both testes be fully descended into the scrotum. Data including age, breed, intraoperative and postoperative complications, medical treatments, and duration of hospitalization were collected retrospectively from medical records.

### 2.2. Preoperative Preparation

Feed was withheld for 12 h prior to surgery. For patients not fully vaccinated upon admission, a tetanus toxoid vaccine was administered along with prophylactic antimicrobials—penicillin G procaine (22,000 IU/kg IM) and gentamicin sulfate (6.6 mg/kg IV)—as well as flunixin meglumine (1.1 mg/kg IV) for anti-inflammatory support.

### 2.3. Anesthesia

General anesthesia was induced using intravenous xylazine hydrochloride (1.1 mg/kg), followed by diazepam (0.05 mg/kg) and ketamine hydrochloride (2.2 mg/kg). Maintenance of anesthesia was achieved with isoflurane delivered via a semi-closed circle system, using positive-pressure ventilation through a cuffed endotracheal tube. All animals were placed in dorsal recumbency within a dedicated surgical suite.

### 2.4. Surgical Technique

All procedures were carried out using a modified semi-closed castration technique. A urinary catheter was inserted into the urethra, absorbent material was placed in the preputial cavity, and the preputial orifice was closed using either a simple continuous suture or Backhaus clamps. The skin over the scrotum, prepuce, caudal abdominal wall, and medial thighs was clipped and aseptically prepared using alternating applications of chlorhexidine gluconate and alcohol; surgical draping was not applied. Local anesthesia was achieved by injecting 10–20 mL of 2% lidocaine directly into each testicle using an 18-gauge with a 3.8 cm needle. After five minutes of skin scrubbing, the anesthetic was allowed to diffuse proximally into the spermatic cord while skin disinfection was completed. A 4–6 cm skin incision was made with a No. 21 scalpel blade over the palpable external inguinal ring. The incision was extended through the tunica dartos and scrotal fascia, taking care not to incise the parietal vaginal tunic. Prior to skin incision, the inguinal ring was palpated to determine its exact location. Once exposed, the fascia underlying the incision was divided and gently widened using digital blunt dissection. Connective tissue was separated to expose the parietal vaginal tunic and the cremaster muscle while carefully preserving the major branches of the external pudendal vein. Blunt digital dissection was employed to circumscribe the spermatic cord. The testicle was mobilized from the scrotum and guided cranially toward the inguinal incision. Gentle traction on the spermatic cord allowed for the exposure of the testicle, still enclosed within its vaginal tunic, through the inguinal wound (Figure 1).

After bluntly dissecting the vaginal tunic from the loose fascia, a 5–8 cm-long window was made through the vaginal tunic with the scalpel blade down to the testicle in its most ventral portion. The testicle was then gently squeezed out of it. After the inspection of the vaginal tunic content, with the use of Mayo scissors, the ligament of the tail of epididymis was severed and the mesorchium was separated from the vaginal tunic as proximally as possible (Figure 2).

Bleeding vessels in the mesorchium were cauterized using a bipolar electrosurgical unit. A giant [10] knot with either Vicryl or PDS II 0 USP was placed as proximally as possible on the vascular bundle (Figure 3).

An emasculator was placed on the vascular bound just distal to the ligature and closed. If hemorrhage was observed, it was controlled by the temporary application of hemostatic forceps. After checking the stump for intraoperative hemorrhage, it was released into the vaginal process, the vaginal tunic was stretched distally with allis forceps, and a giant knot [11] was placed on the vaginal tunic as proximally as possible. Then, the vaginal tunic was emasculated just distal to the ligature (Figure 4).

The subcutaneous tissue over the inguinal ring was closed with Monocryl 2-0 USP simple continuous suture. The skin was closed with a simple continuous or intradermal suture with Monocryl 3-0 USP (Figure 5).

Finally, the Backhaus clamp or continuous suture on the prepuce were removed, and the urinary catheter was checked for position and urine production, before being removed. Equids were closely monitored and allowed to recover from anesthesia.

### 2.5. Postoperative Care and Outcome

A three-day postoperative course of antimicrobials (penicillin G procaine (22,000 IU/kg [10,000 IU/lb]) and Dihydrostreptomycin IM) and phenylbutazone (2.2 mg/kg q24h, PO) was administered to all horses. Equids were evaluated by the authors. Complications were considered only those that required a change in the postoperative plan [12]. In particular, pyrexia was considered a postoperative complication if the rectal temperature was above 39 °C or the duration was >48 h. Swelling was considered a complication only if the attendance of a veterinarian was requested by the owners in the field or if treatment was needed on top of the postoperative treatment described above. Owners were instructed to monitor for any adverse events during the recovery period and to prevent contact with female conspecifics for a minimum of one-month post-orchiectomy. A minimum of 90 days following the procedure, a designated investigator conducted structured telephone follow-ups. These assessments aimed to gather data on delayed-onset clinical issues, wound healing disturbances, and the patient’s ability to resume its prior performance capacity or designated functional role.

## 3. Results

A total of 85 equids were included in this study. The population consisted of 19 Standardbreds, 3 Thoroughbreds, 16 American Quarter Horses, 14 Ponies and 31 warmblood-type saddle horses. Two were donkeys. The age of animals ranged from 18 months to 17 years, with median age of 6.3 years. Median total anesthesia and surgery times were 95 min (range, 60 to 115 min) and 60 min (range, 45 to 95 min), respectively. No intraoperative or anesthetic recovery complications were observed. Two patients—one Standardbred and one Thoroughbred—developed postoperative scrotal swelling, which was resolved with the administration of NSAIDs. No cases of hemorrhage, pyrexia, surgical site infection, or other complications were recorded. Postoperative physical examinations (in all equids) did not reveal any additional abnormalities. All equids had follow-up information available. The overall complication rate was 2 out of 85 (2.3%). All animals without postoperative complications were gradually reintroduced to work or routine activity starting two days post-surgery. All surgical wounds were healed by primary intention, and no further complications were reported by the owners at follow-up assessments conducted at least 3 months postoperatively (range: 3 to 123 months).

## 4. Discussion

This study describes and evaluates a newly developed modified semi-closed castration technique in equids. The overall complication rate observed in this cohort (2 out of 85; 2.35%) was notably low and compares favorably with previously published data, highlighting the potential benefits of the proposed method. Reported complication rates for equine castration vary significantly, ranging from 2% to as high as 48%, depending on the surgical technique employed, perioperative protocols, and environmental conditions under which the procedure is performed [1,7,13]. The inguinal approach adopted in this study appears to offer several advantages when compared to the more commonly used scrotal techniques typically used in field settings. This method facilitates improved aseptic conditions and has been associated with reduced postoperative inflammatory responses, as evidenced by decreased Serum Amyloid A concentrations, as previously reported by Riemersma et al. [6]. The semi-closed nature of the technique—allowing for the initial inspection of the vaginal tunic, followed by its secure closure—offers clear benefits over both traditional open and closed methods. This configuration lowers the risk of postoperative infection and evisceration compared to open approaches, which leave the tunic open to the external environment [1], while still permitting thorough inspection of the tunic’s contents. Additionally, this technique minimizes the volume of tissue engaged by the emasculator or included within the ligatures, representing a notable advantage over fully closed techniques [3,4]. In comparison to previously described semi-closed methods [5,9,10], the current approach introduces further refinements. Notably, it allows for more proximal closure of the vaginal tunic, which may further reduce the risk of postoperative herniation and evisceration. Unlike the techniques reported by Kummer et al. and Petrizzi et al. [9,10], this method facilitates the safe and direct examination of the tunic’s internal structures without posing a risk of inadvertently perforating intestinal loops—an important consideration in animals presenting with inguinal hernia. Moreover, in comparison to the technique described by Crosa et al. [5], the present technique involves reduced tissue manipulation and a reduced number of procedural steps. These features may contribute to a more rapid recovery and a decreased incidence of postoperative complications, as reflected in the swift return to normal activity observed in the study population.

We defined the described technique as a semi-closed technique. Current classifications of castration techniques present some ambiguity, primarily focusing on whether the vaginal tunic is removed [2] or on the status of skin closure [7]. However, these definitions inadequately reflect the critical surgical objective of preventing postoperative complications, particularly herniation and evisceration. We propose a refined classification system based explicitly on the status of the vaginal tunic at the conclusion of the procedure:Closed: The vaginal process remains intact throughout the procedure and is not opened;Open: The vaginal process is opened and remains open at the conclusion of the procedure;Semi-closed: The vaginal process is initially opened for inspection or manipulation and then securely closed at the end of the surgery.

This refined definition effectively encompasses techniques described by Kummer et al. [9], Petrizzi et al. [10] and Crosa et al. [5], as well as the open and semi-closed techniques [2] emphasizing that what truly determines the efficacy of preventing postoperative herniation and evisceration is whether the vaginal tunic remains closed postoperatively, rather than its removal during surgery.

Further, it also encompasses our technique, thus allowing for better communication among surgeons and proper comparison among techniques. Future studies should investigate whether the suture material and the closure of the external inguinal ring can cause postoperative adhesions or discomfort in walking with the hind limbs. Furthermore, a prospective randomized study could be conducted for comparing all surgical techniques.

## 5. Conclusions

In conclusion, our modified semi-closed inguinal castration technique demonstrates a reduction in postoperative complications compared to those reported in the literature for other techniques and approaches [1,3,5,7,9] Its integration into clinical practice could lead to improved surgical outcomes, reduced complication rates, and enhanced recovery profiles for equine patients undergoing castration.

## Figures and Tables

**Figure 1 vetsci-12-00720-f001:**
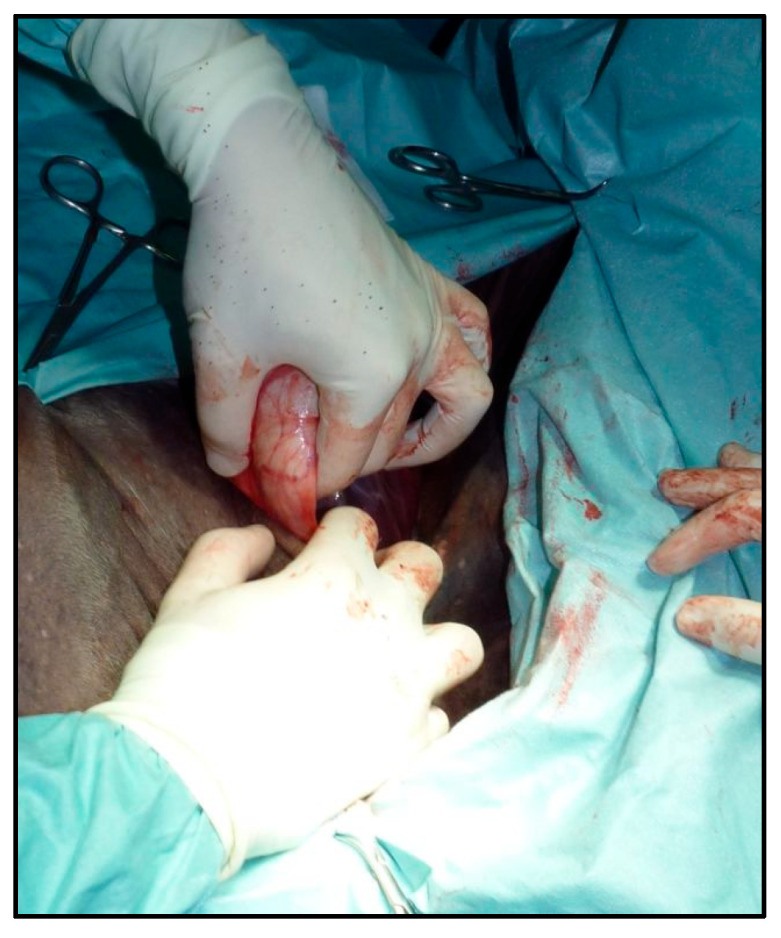
The pushing of the testicle from the scrotum towards the inguinal incision. By gently pulling on the cord, the testicle, enclosed by the parietal tunic, is exposed through the incision (Figure 1).

**Figure 2 vetsci-12-00720-f002:**
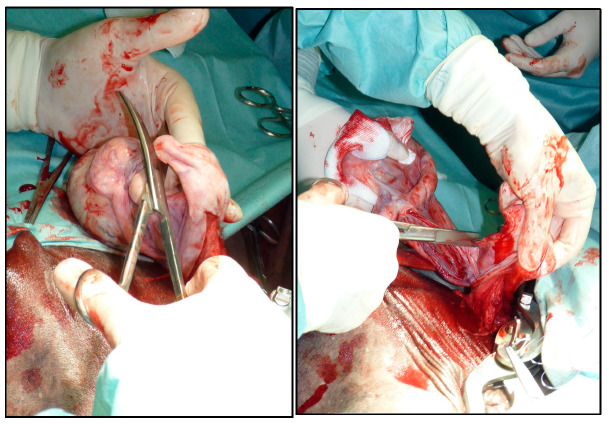
The separation of the mesorchium and cutting of the ligament of the tail of the epididymis (**left**) from the tunica vaginalis as proximally as possible (**right**).

**Figure 3 vetsci-12-00720-f003:**
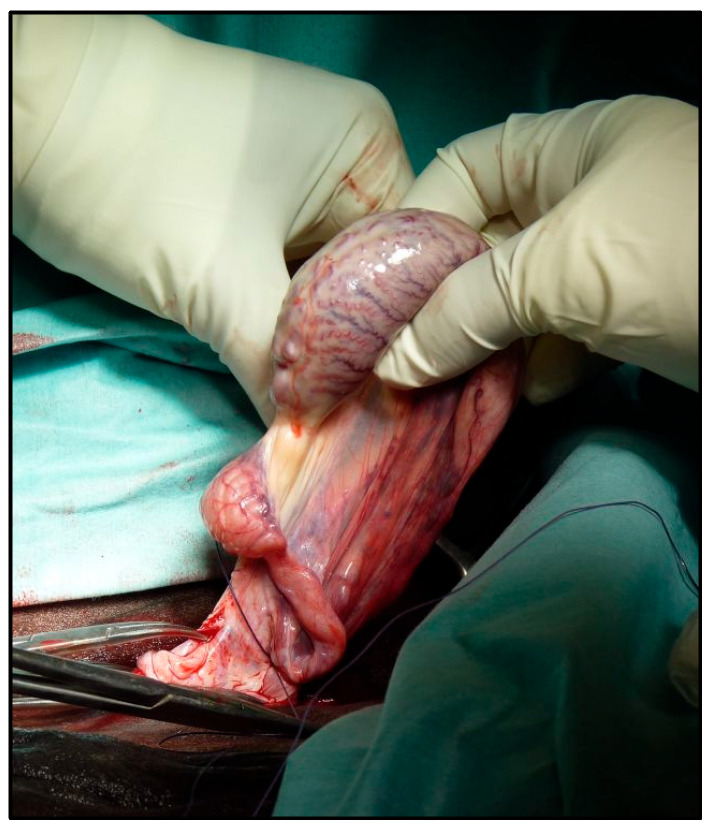
The application of a ligature around the blood vessels.

**Figure 4 vetsci-12-00720-f004:**
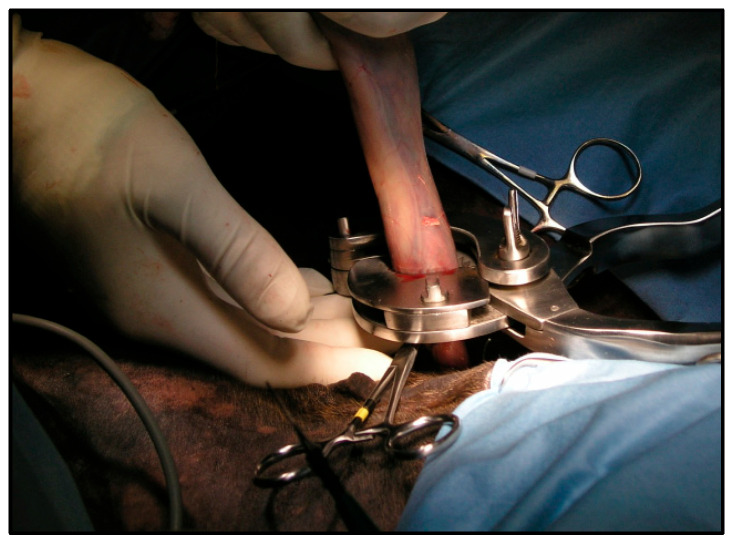
The application of the emasculator over the vaginal tunic.

**Figure 5 vetsci-12-00720-f005:**
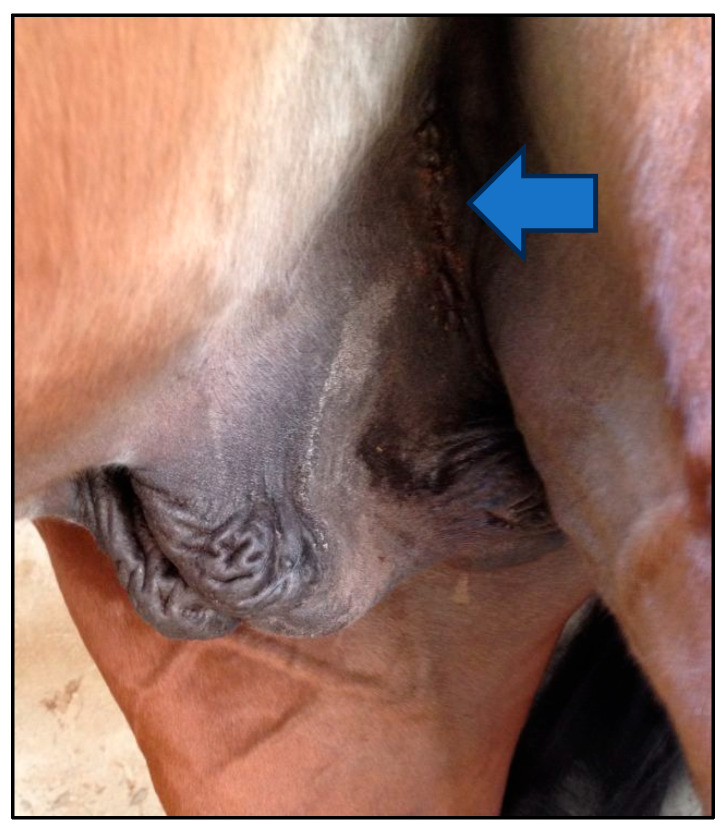
The final aspect of the surgical wound.

## Data Availability

The authors are available to share information and data to support the reported results.

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
