# Peer review of "Description and Complications of a New Modified Semi-Closed Castration Technique in Horses"

_vetsci, 2025, doi:10.3390/vetsci12080720_

Round 1
Reviewer 1 Report
Comments and Suggestions for Authors
The article sounds like a technical note, not a research article.
Abstract
- Too similar to simple summary
- Rewrite it, Abstract should be more scientific and include brief introduction, aims, methods, results and conclusion.
Introduction
- Author has done a good job with introduction, well written, already include background, current limitation and objective
Methods
- Lacking many important detail
- This study should be approved by the animal ethics committee, do you have a certificate? Because it involves using animals for scientific study.
- Author did not mention about number of animals, should be in the method section
o All figures need to be rewrite due to it is too short and insufficient details, remove repeated word “figure”
o Figure 2 author did not explain the difference between left and right images
Results
- Too short to be useful, and cannot assess the quality of research articles, please write more details.
- Maybe show the difference between different breeds and spp.
- Show more data in tables
Discussion
- Without sufficient detail of results, it is difficult to adjust the discussion
- Since you did not directly compare the new and old protocol at the same experiment, how can you state, the new technique is superior to other methods or just speculation, this need to be included in the limitation of this study
- You should not propose something you did not do, for example the classification, you may mention it as limitation
Author Response
REVIEWER 1
The article sounds like a technical note, not a research article
Abstract
R1.1 Too similar to simple summary. Rewrite it, Abstract should be more scientific and include brief introduction, aims, methods, results and conclusion.
R1.1 Rewrited line 35-55
Introduction
Author has done a good job with introduction, well written, already include background, current limitation and objective
Methods
R1.2 Lacking many important detail. This study should be approved by the animal ethics committee, do you have a certificate? Because it involves using animals for scientific study.
R 1.2 Informed consent from the owner was obtained prior to surgery. Supervisory committee approval was not required because the surgery was performed routinely by the authors and the treatment was considered to be in accordance with the standard of care of our hospital. Line 96-98
R1.3 Author did not mention about number of animals, should be in the method section
R1.3 Inserted. See text line 94
R.1.4 All figures need to be rewrite due to it is too short and insufficient details, remove repeated word “figure”
R1.4 Changed. See text line 137-140; 148-150; 155; 163; 168
R1.5 Figure 2 author did not explain the difference between left and right images
R1.5 Inserted. See text 148-150
Results
R1.6 Too short to be useful, and cannot assess the quality of research articles, please write more details
R1.6 As per suggestion of the reviewer, the manuscript type has been changed to “technical note”, also because of this issue. Nevertheless we inserted some relevant details in the results section
R1.7 Maybe show the difference between different breeds and spp.
R1.7 Inserted. See text line 194-196
R1.8 Show more data in tables
R1.8 We would like to have the Editor opinion on this matter. We believe a table will be redundant with what reported in the text.
Discussion
Without sufficient detail of results, it is difficult to adjust the discussion
R1.9 Since you did not directly compare the new and old protocol at the same experiment, how can you state, the new technique is superior to other methods or just speculation, this need to be included in the limitation of this study. You should not propose something you did not do, for example the classification, you may mention it as limitation
R1.9 Changed. See text line 245-247
ùREVIEWER 2
Dear Authors:
the work describes well, with great attention to detail, the execution of the surgical technique.
R2.1 The semi-closed definition in my opinion is not correct; the technique is open with closure and healing by first intention. After surgery, no structure remains open and no tissue is not closed. At the end of the procedure all the planes are sutured and closed. The surgical technique has been used in the last 20 years with different techniques of hemostasis and ligation of the spermatic cord. The use of this technique was initially described for entire adult male horses and stallions, with larger inguinal ring and thicker spermatic cord potentially increasing the risk of hernia and disembowelment
During the reading, I do not highlight a new technique, but the correct description of the surgical technique that is already widely used.
R2.1 we respectfully disagree with the reviewer. The technique described is different from the previously described techniques. We agree that the inguinal approach is the same as in Kummer et al, but the orchiectomy technique is not. And is not the same as described by Crosa et. al, and from those previously (Schumacher et al)
The confusion may be caused from what we report in the discussion, i-e- the lack of a classification based on the complications that different surgical techniques aim to reduce, hernia and evisceration.
In fact, one technique have been reported with a higher rate of evisceration and closure of the vaginal tunic in semi-closed and closed techniques aims to reduce this risk.
For these reasons we propose a refined classification system based explicitly on the state of the tunica vaginalis at the end of the procedure.
- Closed: The processus vaginalis remains intact throughout the procedure and is not opened.
- Open: The processus vaginalis is opened and remains open at the end of the procedure.
- Semi-closed: The processus vaginalis is initially opened for inspection or manipulation and then closed tightly at the end of the procedure.
This classification encompasses and classifies all the reported orchiectomy techniques to date.
The study from Petrizzi et al has been included in the references. In that paper it is not described the same technique as reported in our study. It is a semi-closed technique, in the meaning that the tunic is initially incised and then closed but the difference lies in the fact that with that technique, herniation into the vaginal tunic is still possible. In the technique we describe, the vaginal tunic is brought out of the skin incision, incised and removed after been ligated as proximally as possible. With this method, herniation is possible only in a very limited portion of vaginal tunic that can host a very little amount of intestine.
Closure of the skin or second intention healing of it is not related to reducing these complications but solely to the contamination of the wound that differs between field and hospital setting when performing the surgery.
R2.3 The surgical procedure can in any case present complications related to the suture materials used, and suturing planes in the future could lead to adhesions near the external inguinal ring and discomfort in the hind limb gait.
R2.3 Inserted. See text
Reference
R2.4 Kummer, M., Gygax, D., Jackson, M., Bettschart-Wolfensberger, R., Fürst, A. Results and complications of a novel technique for primary castration with an inguinal approach in horses. Equine Vet J 2009, 41(6), 547-551. 267
R 2.4 This reference was already cited in the text
R2.5 Petrizzi, L., Fürst, A., Lischer, C. Clinical evaluation of a vessel sealing device (LigaSureTM) for haemostasis of the testicular vessels for castration of stallions using an inguinal approach and primary closure (2006) Wiener Tierarztliche Monatsschrift, 93 (5-6), pp. 120-126
R2.5 Inserted. See text

Reviewer 2 Report
Comments and Suggestions for Authors
Dear Authors:
the work describes well, with great attention to detail, the execution of the surgical technique.
The semi-closed definition in my opinion is not correct; the technique is open with closure and healing by first intention
After surgery, no structure remains open and no tissue is not closed
At the end of the procedure all the planes are sutured and closed
The surgical technique has been used in the last 20 years with different techniques of hemostasis and ligation of the spermatic cord.
The use of this technique was initially described for entire adult male horses and stallions, with larger inguinal ring and thicker spermatic cord potentially increasing the risk of hernia and disembowelment
During the reading, I do not highlight a new technique, but the correct description of the surgical technique that is already widely used.
The surgical procedure can in any case present complications related to the suture materials used, and suturing planes in the future could lead to adhesions near the external inguinal ring and discomfort in the hind limb gait.
Reference
Kummer, M., Gygax, D., Jackson, M., Bettschart-Wolfensberger, R., Fürst, A. Results and complications of a novel technique for primary castration with an inguinal approach in horses. Equine Vet J 2009, 41(6), 547-551. 267
Petrizzi, L., Fürst, A., Lischer, C. Clinical evaluation of a vessel sealing device (LigaSureTM) for haemostasis of the testicular vessels for castration of stallions using an inguinal approach and primary closure (2006) Wiener Tierarztliche Monatsschrift, 93 (5-6), pp. 120-126. Cited 6 times.
Petrizzi L., Fuerst A., Varasano V., Bettschart Wolfenberger R., Lischer C.(2007): Castrazione dello stallone con approccio inguinale e sutura della breccia operatoria. Atti 13th Congresso S.I.V.E., 178.
Author Response
REVIEWER 2
Dear Authors:
the work describes well, with great attention to detail, the execution of the surgical technique.
R2.1 The semi-closed definition in my opinion is not correct; the technique is open with closure and healing by first intention. After surgery, no structure remains open and no tissue is not closed. At the end of the procedure all the planes are sutured and closed. The surgical technique has been used in the last 20 years with different techniques of hemostasis and ligation of the spermatic cord. The use of this technique was initially described for entire adult male horses and stallions, with larger inguinal ring and thicker spermatic cord potentially increasing the risk of hernia and disembowelment
During the reading, I do not highlight a new technique, but the correct description of the surgical technique that is already widely used.
R2.1 we respectfully disagree with the reviewer. The technique described is different from the previously described techniques. We agree that the inguinal approach is the same as in Kummer et al, but the orchiectomy technique is not. And is not the same as described by Crosa et. al, and from those previously (Schumacher et al)
The confusion may be caused from what we report in the discussion, i-e- the lack of a classification based on the complications that different surgical techniques aim to reduce, hernia and evisceration.
In fact, one technique have been reported with a higher rate of evisceration and closure of the vaginal tunic in semi-closed and closed techniques aims to reduce this risk.
For these reasons we propose a refined classification system based explicitly on the state of the tunica vaginalis at the end of the procedure.
- Closed: The processus vaginalis remains intact throughout the procedure and is not opened.
- Open: The processus vaginalis is opened and remains open at the end of the procedure.
- Semi-closed: The processus vaginalis is initially opened for inspection or manipulation and then closed tightly at the end of the procedure.
This classification encompasses and classifies all the reported orchiectomy techniques to date.
The study from Petrizzi et al has been included in the references. In that paper it is not described the same technique as reported in our study. It is a semi-closed technique, in the meaning that the tunic is initially incised and then closed but the difference lies in the fact that with that technique, herniation into the vaginal tunic is still possible. In the technique we describe, the vaginal tunic is brought out of the skin incision, incised and removed after been ligated as proximally as possible. With this method, herniation is possible only in a very limited portion of vaginal tunic that can host a very little amount of intestine.
Closure of the skin or second intention healing of it is not related to reducing these complications but solely to the contamination of the wound that differs between field and hospital setting when performing the surgery.
R2.3 The surgical procedure can in any case present complications related to the suture materials used, and suturing planes in the future could lead to adhesions near the external inguinal ring and discomfort in the hind limb gait.
R2.3 Inserted. See text
Reference
R2.4 Kummer, M., Gygax, D., Jackson, M., Bettschart-Wolfensberger, R., Fürst, A. Results and complications of a novel technique for primary castration with an inguinal approach in horses. Equine Vet J 2009, 41(6), 547-551. 267
R 2.4 This reference was already cited in the text
R2.5 Petrizzi, L., Fürst, A., Lischer, C. Clinical evaluation of a vessel sealing device (LigaSureTM) for haemostasis of the testicular vessels for castration of stallions using an inguinal approach and primary closure (2006) Wiener Tierarztliche Monatsschrift, 93 (5-6), pp. 120-126
R2.5 Inserted. See text

Round 2
Reviewer 1 Report
Comments and Suggestions for Authors
I am happy with this version of the manuscript, but it should be published as a technical note because it is not suitable for a research article. This manuscript could be useful for improve the surgical method in equid species.